# Fiber-specific structural properties relate to reading skills in children and adolescents

Steven Lee Meisler[1]*[†], John DE Gabrieli[2†]

[1]Program in Speech and Hearing Bioscience and Technology, Harvard Medical School, Boston, United States; [2]McGovern Institute for Brain Research, Cambridge, United States

**Abstract** Recent studies suggest that the cross-sectional relationship between reading skills and white matter microstructure, as indexed by fractional anisotropy, is not as robust as previously thought. Fixel-based analyses yield fiber-specific micro- and macrostructural measures, overcoming several shortcomings of the traditional diffusion tensor model. We ran a whole-brain analysis investigating whether the product of fiber density and cross-section (FDC) related to single-word reading skills in a large, open, quality-controlled dataset of 983 children and adolescents ages 6–18. We also compared FDC between participants with (n = 102) and without (n = 570) reading disabilities. We found that FDC positively related to reading skills throughout the brain, especially in left temporo-parietal and cerebellar white matter, but did not differ between reading proficiency groups. Exploratory analyses revealed that among metrics from other diffusion models – diffusion tensor imaging, diffusion kurtosis imaging, and neurite orientation dispersion and density imaging – only the orientation dispersion and neurite density indexes from NODDI were associated (inversely) with reading skills. The present findings further support the importance of left-hemisphere dorsal temporoparietal white matter tracts in reading. Additionally, these results suggest that future DWI studies of reading and dyslexia should be designed to benefit from advanced diffusion models, include cerebellar coverage, and consider continuous analyses that account for individual differences in reading skill.

**\*For correspondence:**
smeisler@g.harvard.edu

**Present address:** [†]Department of Brain & Cognitive Sciences, Massachusetts Institute of Technology, Cambridge, United States

**Competing interest:** The authors declare that no competing interests exist.

## Editor's evaluation

This valuable study investigates the association between fixel-based white matter measures and reading for the first time. In a large sample of participants ranging from 6-18 years of age, a convincing association between intra-axonal volume and single-word reading abilities are reported. This work will be of interest to a wide readership.

## Introduction

Many research efforts spanning multiple neuroimaging modalities have sought to yield insights into the neural bases of reading ability and disability (*Vandermosten et al., 2012*; *Landi et al., 2013*; *Richlan et al., 2013*). Among these studies are those that employ diffusion-weighted imaging (DWI) to study the properties of anatomical connections in the brain. The most commonly reported measure of white matter microstructure is fractional anisotropy (FA). FA is a metric derived from the diffusion tensor imaging (DTI) model (*Basser et al., 1994*) that quantifies the degree to which water diffusion is directionally restricted in each voxel (*Hagmann et al., 2006*; *Basser and Pierpaoli, 2011*). FA is high in white matter compared with gray matter and cerebrospinal fluid (CSF) due to preferential water movement along the axis of axons. Studies of white matter microstructural properties' relationships to

reading skill have primarily used FA (for overviews, see *Ben et al., 2007*; *Vandermosten et al., 2012*; *Moreau et al., 2018*; *Meisler and Gabrieli, 2022*). However, several factors confound the ability to draw meaningful interpretations from FA results (*Farquharson et al., 2013*; *Riffert et al., 2014*). As a metric defined on the voxel-level, FA is prone to partial volume effects, manifesting as reduced FA in regions where white matter borders gray matter or CSF (*Vos et al., 2011*). Due to the limited degrees of freedom in the tensor model, FA is artificially lower in regions of crossing fibers, affecting up to 90% of white matter voxels (*Behrens et al., 2007*; *Jeurissen et al., 2013*). In addition to sensitivity to myelination, FA also tends to covary with other elements such as axonal diameter, density, permeability, and coherence (*Beaulieu, 2009*; *Johansen-Berg and Behrens, 2013*; *Shemesh, 2018*; *Friedrich et al., 2020*; *Lazari and Lipp, 2021*), and information from DTI alone is not sufficient to gauge the individual contributions of these features. Thus, FA has often been reduced to a nonspecific (and arguably inappropriate; see *Jones et al., 2013*) term, 'white matter integrity'.

Early cross-sectional studies of FA and reading skills seemed to converge towards a consensus of greater FA relating to better reading ability, particularly in left temporoparietal white matter tracts that connect neocortical regions known to be important for language, such as the arcuate fasciculus (AF) and superior longitudinal fasciculus (SLF) (*Klingberg et al., 2000*; *Ben et al., 2007*; *Vandermosten et al., 2012*). As tract segmentation algorithms became more robust and widely used, subsequent studies, empowered to address tract-specific hypotheses, began describing a range of results. These included significant FA-reading relationships in different areas, such as commissural (*Frye et al., 2008*; *Lebel et al., 2013*), cerebellar (*Travis et al., 2015*; *Bruckert et al., 2020*), and right-lateralized bundles (*Horowitz-Kraus et al., 2015*), as well as regions where higher FA was associated with worse reading skills (*Carter et al., 2009*; *Frye et al., 2011*; *Christodoulou et al., 2017*). The inconsistency in past results is potentially driven by a variety of factors such as publication bias (*Begg, 1994*), small participant cohorts, inhomogeneous acquisition parameters, different covariates and reading measures, variation in age groups, and different processing techniques (*Moreau et al., 2018*; *Ramus et al., 2018*; *Schilling et al., 2021a*; *Schilling et al., 2021b*). Few studies have sought to resolve these inconclusive results. A meta-analysis of whole-brain voxel-based studies found no regions where FA either varied with reading ability or was reduced in dyslexic compared with typically reading, individuals (*Moreau et al., 2018*). *Geeraert et al., 2020* used principal component analysis to draw out white matter structural indices from several scalar maps, including metrics from DTI (such as FA) and neurite orientation dispersion and density imaging (NODDI; *Zhang et al., 2012*), and found that variance in these measures was driven by age-related development, but not reading. Three large-scale cross-sectional studies using publicly available datasets found largely null associations between FA and reading skills in several tracts (*Koirala et al., 2021*; *Meisler and Gabrieli, 2022*; *Roy et al., 2022*).

Despite the mixed empirical findings relating FA to reading skill, it is reasonable to hypothesize that there ought to be such a brain structure–behavior correlate of reading ability. Reading involves the functioning of a widely distributed brain network (*Cattinelli et al., 2013*; *Wandell and Yeatman, 2013*; *Murphy et al., 2019*), and white matter tracts are conduits for information sent within this network (*Ben et al., 2007*). Lesion-mapping analyses (*Wang et al., 2020*; *Li et al., 2021*) and clinical case studies (*Epelbaum et al., 2008*; *Rauschecker et al., 2009*) have demonstrated that white matter connections, primarily in the left hemisphere, are necessary for reading. Since white matter exhibits learning-driven plasticity and can also modulate neuronal firing patterns (*Fields, 2015*; *Xin and Chan, 2020*), one may expect that functional variation, such as differences in reading ability, may be reflected by *some* white matter structural property (*Ramus et al., 2018*; *Protopapas and Parrila, 2018*; *Protopapas and Parrila, 2019*). The largely null findings in higher-powered meta-analyses (*Moreau et al., 2018*) and large-scale studies (*Koirala et al., 2021*; *Meisler and Gabrieli, 2022*; *Roy et al., 2022*) suggest that FA is not a specific enough metric to effectively capture this relationship in cross-sectional designs (however, see *Van Der Auwera et al., 2021* and *Roy et al., 2022* for evidence that FA tracks individual longitudinal trajectories in reading achievement).

More advanced diffusion models have yielded metrics that better reflect variance in reading skills. *Sihvonen et al., 2021* found that connectometry from quantitative anisotropy modeling (*Yeh et al., 2013*) in multiple pathways covaried with better reading skill independently from phonological abilities. Quantitative anisotropy is less prone to artifacts from partial volume effects and crossing fibers than FA (*Yeh et al., 2016*). *Zhao et al., 2016* found that more right-sided laterality of hinderance-modulated orientation anisotropy (HMOA; *Dell'Acqua et al., 2013*) in the SLF and inferior

frontal-occipital fasciculus was related to worse reading skills. *Koirala et al., 2021* reconstructed multiple diffusion models in children and concluded that lower orientation dispersion and neurite density indices from NODDI modeling related to better reading abilities in several bilateral tracts, while FA was not associated with reading. Although not a DWI sequence, myelin water imaging (MWI) studies have suggested both positive (*Beaulieu et al., 2020*) and negative (*Economou et al., 2022*) associations of myelination with reading skill in children. *Economou et al., 2022* also replicated null associations between FA and reading ability in their experimental cohort. These results collectively suggest that studies of reading (and perhaps other cognitive domains; see *Lazari et al., 2021*) should begin to move beyond traditional DTI modeling. However, NODDI metrics, being a voxel-level metric, cannot ascribe properties to particular fiber populations if multiple exist in a voxel. MWI acquisitions, while showing higher specificity to variation in myelin, tend to have relatively long scan times (*Alonso-Ortiz et al., 2015*) one would also still need to collect a DWI scan if one wanted to associate MWI metrics with fiber bundles and properly account for MWI variation due to fiber orientations (*Birkl et al., 2021*). Collecting all of these data in children and clinical populations is challenging and not always practical.

Subsequently, a DWI analytical paradigm was introduced that performs statistical inferences on 'fixels,' or individual fiber populations within voxels, using a set of three fixel-derived metrics: fiber density (FD), fiber cross-section (FC), and their product (FDC) (*Raffelt et al., 2015*). This framework is enabled by constrained spherical deconvolution (CSD) (*Tournier et al., 2007*), a data-driven approach for resolving fiber orientation distributions (FODs) even in the presence of crossing fibers. Unlike other fiber-specific metrics, such as quantitative anisotropy, fixel-based analyses (FBA) can yield distinct micro- and macrostructural components, and these can be studied on a fixel-by-fixel basis, affording increased spatial specificity. FD is a microstructural measure that reflects the intra-axonal volume fraction (*Raffelt et al., 2012b*; *Genc et al., 2020*), while FC is a macrostructural measure related to the cross-sectional area of fiber bundles (*Raffelt et al., 2012b*). The product of FD and FC, or FDC, is therefore related to the total estimated intra-axonal volume and is sensitive to both white matter micro- and macrostructure. Increased intra-axonal volume may reflect either an increased number of axons in a given area or the presence of wider axons (or some combination thereof), although conventional DWI alone may not be able to resolve the respective contributions of these two possibilities. Wider axons conduct action potentials more quickly and can fire more often at their terminals (*Perge et al., 2012*). Thus, FDC is thought to more closely relate to the conductive capacity of white matter (*Raffelt et al., 2017b*).

In addition to enabling investigations of these more specific fixel-derived metrics, FBA present several additional advantages compared to traditional FA whole-brain approaches (*Dhollander et al., 2021a*). Since FBAs operate on the level of fixels, and fixels are generated from FODs in white matter, FBAs are by nature restricted to white matter, thus mitigating the effects of multiple comparison correction from redundant regions in other neural compartments. Spatial smoothing in FBAs is performed within local neighborhoods of white matter bundles informed by fixel connectivity (*Raffelt et al., 2015*). Thus, the signal in a given fixel is not influenced by different tissue classes or other fiber populations, in contrast to traditional voxel-based spatial smoothing, which operates more indiscriminately.

FBAs have been quickly adopted and used to investigate several clinical and developmental populations (reviewed in *Dhollander et al., 2021a*). However, they have not yet been used to examine reading abilities. With the increased specificity of FBAs, this approach might reveal fiber-specific biomarkers that are more sensitive to variation in reading abilities than FA or other tensor-derived metrics, providing valuable insights into the neural basis of literacy. In this study (*Figure 1*), we examined the relationship between single-word reading skill and FDC (primary analysis), FD, and FC (secondary analyses) in a pediatric dataset of 983 children and adolescents ages 6–18 from the Healthy Brain Network (HBN) biobank (*Alexander et al., 2017*). We additionally looked for differences in fixel metrics between participants with (n = 102) and without reading disabilities (n = 570) using the criteria based on diagnostic and standardized reading assessments. In a set of exploratory analyses, we tested whether DWI metrics from other models – DTI, diffusion kurtosis imaging (DKI; *Jensen et al., 2005*), and NODDI – were related to reading abilities. In all analyses, we employed generalized additive modeling (GAM) (*Hastie and Tibshirani, 1990*) to more flexibly model age-related variance given the wide age range of participants (*Zhao et al., 2022*; *Bethlehem et al., 2022*).

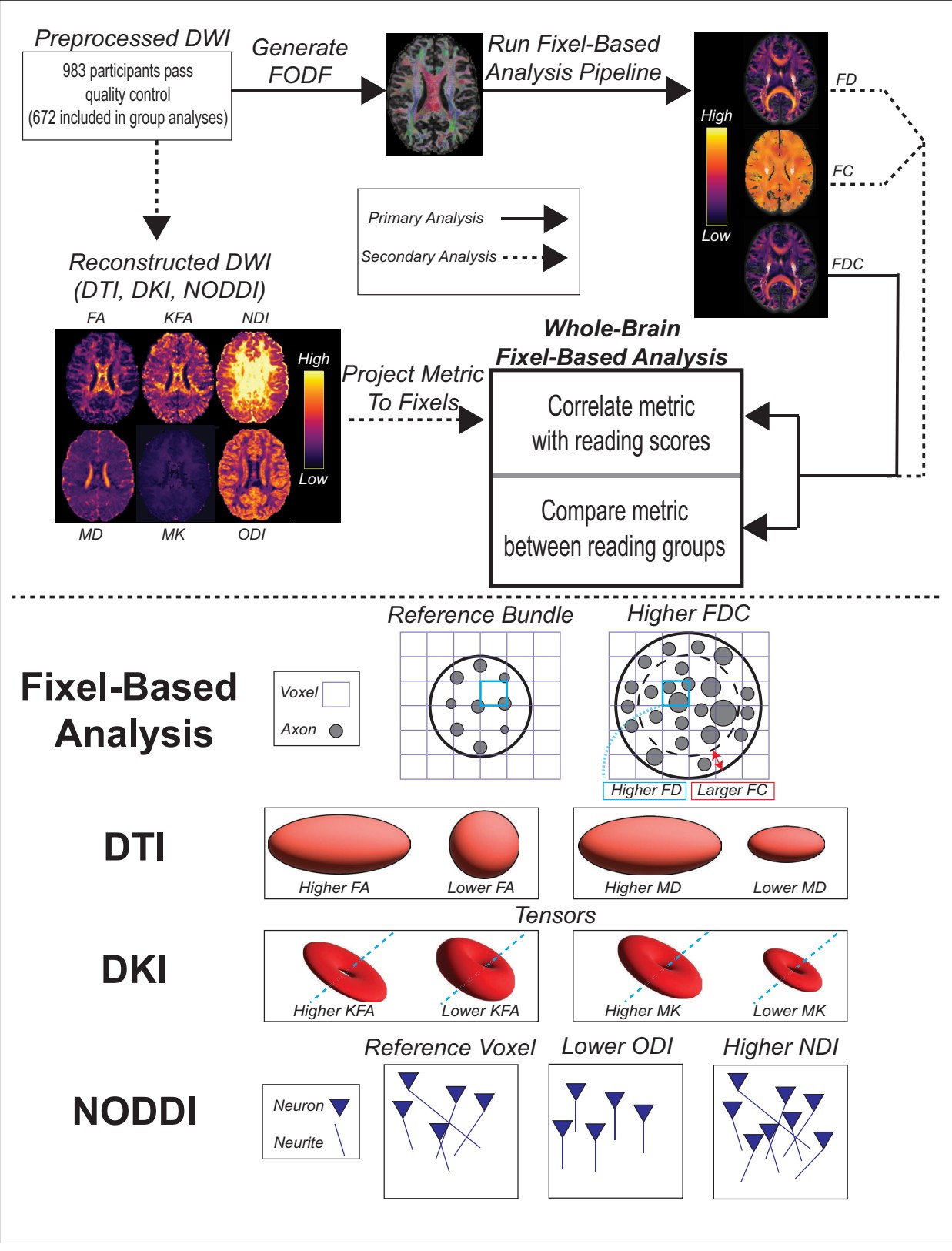

**Figure 1.** Methodological overview of the study. Top: description of primary and secondary analyses. Bottom: schematic depicting interpretations of changes in examined metrics. Depictions of bundles, axons, and neurites are not drawn to scale. DWI, diffusion-weighted imaging; DTI, diffusion tensor imaging; DKI, diffusion kurtosis imaging; NODDI, neurite orientation density and dispersion index; FA, fractional anisotropy; KFA, kurtosis fractional anisotropy; MD, mean diffusivity; MK, mean kurtosis; NDI, neurite density index; ODI, orientation dispersion index; FODF, fiber orientation distribution function; FD, fiber density; FC, fiber cross-section; FDC, fiber density and cross-section product.

**Table 1.** Phenotypic and neuroimaging summary statistics in all participants and within the two reading proficiency groups. 17 and 93 participants were lacking socioeconomic and WISC scores, respectively, and were ignored for the corresponding rows. Values are listed as mean (standard error of the mean). For group comparison effect sizes (right-most column), *p<0.05 and † p<0.001. All *t*-tests were Welch's *t*-tests, and $\chi^2$ tests were used for comparisons of categorical variables.

| Metric | All (n = 983) | TR (n = 570) | RD (n = 102) | Effect size |
|---|---|---|---|---|
| Sex (M/F) | 617/366 | 355/215 | 59/43 | $\Phi = 0.0235$ |
| Age (years) | 11.16 (0.10) | 11.38 (0.14) | 10.56 (0.27) | $d = 0.258*$ |
| Handedness (EHI) | 61.78 (1.58) | 62.19 (2.05) | 62.91 (5.05) | $d = 0.015$ |
| Handedness (L/A/R) | 74/128/781 | 42/66/462 | 8/17/77 | $\Phi = 0.047$ |
| SES (years parental edu.) | 17.63 (0.10) | 18.13 (0.11) | 16.93 (0.32) | $d = 0.429†$ |
| ICV (cm$^3$) | 1540 (5.130) | 1559 (6.735) | 1501 (12.47) | $d = 0.370†$ |
| WISC VSI | 102.08 (0.552) | 105.72 (0.714) | 97.82 (1.497) | $d = 0.494†$ |
| WISC VCI | 104.61 (0.542) | 109.26 (0.658) | 98.18 (1.414) | $d = 0.750†$ |
| TOWRE | 97.93 (0.56) | 109.49 (0.45) | 70.48 (0.80) | $d = 3.74†$ |
| Global FD | 0.285 (6.26e-4) | 0.287 (7.66e-4) | 0.280 (2.53e-3) | $d = 0.337*$ |
| Global log(FC) | 0.050 (2.15e-3) | 0.059 (2.73e-3) | 0.030 (5.92e-3) | $d = 0.455†$ |
| Mean motion (mm) | 0.44 (7.89e-3) | 0.44 (0.01) | 0.44 (0.03) | $d = 4.27e-3$ |
| Quality (Neighbor Corr.) | 0.756 (1.58e-3) | 0.760 (2.08e-3) | 0.745 (5.17e-3) | $d = 0.291*$ |

TR = typically reading group; RD = reading disability group; EHI = Edinburgh Handedness Inventory; SES = socioeconomic status; ICV = intracranial volume; TOWRE = Tests of Word Reading Efficiency composite score, age-normalized; WISC VSI = Wechsler Intelligence Scale for Children visuospatial index, age-normalized; WISC VCI = Wechsler Intelligence Scale for Children verbal comprehension index, age-normalized; FD = fiber density; FC = fiber cross-section. FD and FC are unitless.

We hypothesized that we would see positive associations between FDC and reading abilities, as well as lower FDC among dyslexic readers, in several tracts spanning both hemispheres, but especially the left arcuate fasciculus, left inferior fronto-occipital fasciculus, and cerebellar peduncles, as these tracts yielded significant relationships in multiple studies of advanced diffusion models and reading (*Beaulieu et al., 2020*; *Koirala et al., 2021*; *Sihvonen et al., 2021*; *Economou et al., 2022*). However, since this was the first FBA involving reading skill, and one with considerably high statistical power, we took a more conservative approach and ran a whole-brain FBA. Using tract segmentation, we ascribed locations of significant results to bundles to guide future research efforts.

## Results
### Participant data
The 983 participants who passed all inclusion, exclusion, and quality control criteria (*Table 1*) were divided into a typically reading group (TR; n = 570) and reading disability group (RD; n = 102) based on diagnostic and standardized reading assessments (*Figure 2*; see 'Materials and methods'). A total of 311 participants did not meet the criteria for either group, but were still included in the correlation analyses. The TR group, compared with the RD group, was older and had higher socioeconomic scores, brain volumes, verbal IQ, visuospatial IQ, age-normalized reading scores, globally averaged fixel metrics, and image quality (as indexed by the average neighbor correlation; see *Yeh et al., 2019* for more information on this metric). The groups were matched in sex distribution (although the cohort as a whole was male-skewed), handedness, and average motion (mean framewise displacement). Reading scores and IQs were age-standardized composite indexes from the Tests of Word Reading Efficiency (TOWRE; *Torgesen et al., 1999*) and Wechsler Intelligence Scale for Children (WISC; *Wechsler and Kodama, 1949*), respectively. In total, 17 participants were missing socioeconomic information, and 93 participants did not have WISC scores. Since these variables were not ultimately included in our statistical models, we did not exclude these participants. The relationships between phenotypic and global neuroimaging metrics, and the differences in these measures between

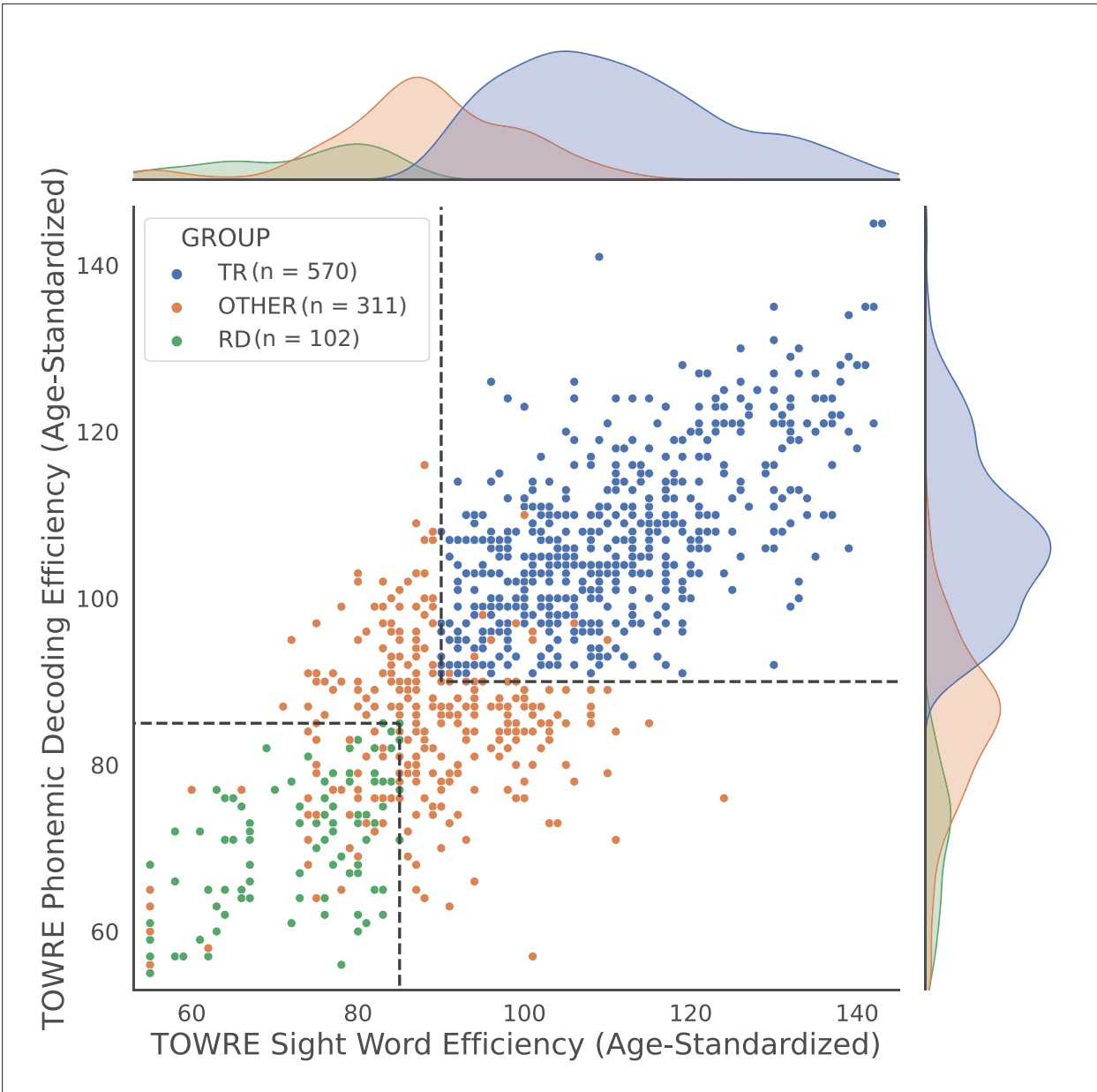

**Figure 2.** Age-standardized TOWRE subscores of all participants. Each dot represents a participant, color-coded by group assignment. Dashed lines mark the score cutoffs for the two reading proficiency groups. Since scores are discrete and not unique, some dots may overlap with each other. Kernel density estimation plots along the perimeter show the distribution of reading scores in each group. TR, typically reading group; RD, reading disability group; TOWRE, Tests of Word Reading Efficiency.

The online version of this article includes the following figure supplement(s) for figure 2:

**Figure supplement 1.** Correlations between continuous phenotypic and neuroimaging variables.

**Figure supplement 2.** ANOVA results for site-wise comparisons between phenotypic and neuroimaging metrics.

scanning sites, can be found in the supplementary materials (*Supplementary file 1*; *Figure 2—figure supplements 1 and 2*).

## Fixel metrics

We ran a whole-brain fixel-based analysis testing whether the product of fiber density and fiber cross-section, or FDC, was associated with raw composite TOWRE scores, controlling for age, sex, intra-cranial volume, image quality, and scanning site. We found widespread bilateral and commissural

regions in which higher FDC was significantly related to better reading abilities ($q_{FDR} < 0.05$; *Figure 3*, *Figure 3—figure supplement 1*). There were no appreciable clusters in which an inverse relationship between FDC and reading skills was observed. Each tract produced by the segmentation software, *TractSeg* (*Wasserthal et al., 2018a*), contained significant fixels (*Table 2*). We defined effect size in each fixel as the difference in adjusted $R^2$ values between the full model and a reduced model without the predictor of interest (e.g., TOWRE scores or group designations). The effect size of significant fixels varied up to a peak value of 0.030. Clusters of fixels with the largest effect sizes ($\Delta R^2_{adj} > 0.028$) were observed in left-hemisphere temporoparietal and cerebellar white matter. These clusters survived at $q_{FDR} < 0.001$ (*Table 2*), which more than accounts for Bonferroni correction across all models described in this study (given $\alpha = 0.05$). Tract segmentation intersections (*Table 2*) revealed that the temporoparietal cluster was most likely associated with the left arcuate fasciculus (AF), superior longitudinal fasciculus (SLF), or middle longitudinal fasciculus (MLF). These tracts overlapped in several areas (*Figure 3—figure supplement 2*). The cerebellar cluster was most likely associated with the left superior cerebellar peduncle (SCP). Homotopic clusters of significant fixels were observed in right-hemisphere temporoparietal and cerebellar white matter, but they reached smaller effect sizes than those in the left hemisphere. Post-hoc exploration of FD and FC revealed diffuse associations of better reading skills with higher FC compared with fewer regions where higher FD was related to better reading (*Figure 3—figure supplement 3*). As expected, highest effect sizes of FDC were achieved in regions where higher FD and FC were both independently related with better reading. We did not find any significant differences in FDC between the TR and RD groups.

Given the wide age range of participants, we also investigated whether the correlation between FDC and TOWRE scores was stable across ages. We ran a smooth bivariate interaction model testing whether there was an interaction between age and TOWRE scores in predicting FDC. Only two trivially small clusters (consisting of one and seven fixels) showed age-related variance in FDC-TOWRE relationships. These small clusters did not intersect with significant fixels from the primary analysis, suggesting that the relationship between FDC and reading skills was stable across ages. In the supplementary materials, we also report the effect size maps of the individual SWE and PDE subscores with FDC (*Figure 3—figure supplement 4*). These maps were qualitatively similar, each notably retaining the peak effect sizes in left temporoparietal and cerebellar regions identified in the primary analysis.

## DTI, DKI, and NODDI analyses

We similarly examined whether metrics from other diffusion models were related to raw TOWRE scores (*Figure 4*). We found that metrics from DTI (FA and mean diffusivity [MD]) and DKI (kurtosis fractional anisotropy [KFA] and mean kurtosis [MK]) did not relate to reading skills. There were a few small areas, primarily in the cerebellum, where the neurite density index (NDI) from NODDI was inversely related to TOWRE skills (max $\Delta R^2_{adj}$ = 0.18). The orientation dispersion index (ODI) from NODDI was also inversely related to reading skills, achieving a max $\Delta R^2_{adj}$ of 0.20. For ODI, the regions of highest effect sizes overlapped with the left temporoparietal and bilateral cerebellar regions that were significant in the primary analysis of FDC. Clusters in neither of the NODDI models survived multiple comparison correction across hypotheses (Bonferroni factor of 12).

## Discussion

In this study, we employed a method to study fiber-specific properties as they relate to single-word reading abilities and disabilities among children and adolescents. We hypothesized that FDC would covary with reading abilities and be lower in dyslexic readers, especially in the left arcuate fasciculus, left inferior fronto-occipital fasciculus, and cerebellum. Unlike our secondary analyses and recent cross-sectional studies that yielded few-to-no regions exhibiting significant FA-reading relationships or group differences in FA (*Moreau et al., 2018*; *Koirala et al., 2021*; *Economou et al., 2022*; *Meisler and Gabrieli, 2022*; *Roy et al., 2022*), we found that higher FDC related to better single-word reading skills throughout the brain. This relationship was stable across ages. However, FDC did not differ between those with and without reading disabilities. Although significant correlations were observed bilaterally, the strongest effect sizes were in the left hemisphere, and especially in temporoparietal and cerebellar white matter. The tracts most likely associated with the regions of strongest correlations were the left-hemisphere AF, SLF, MLF, and SCP.

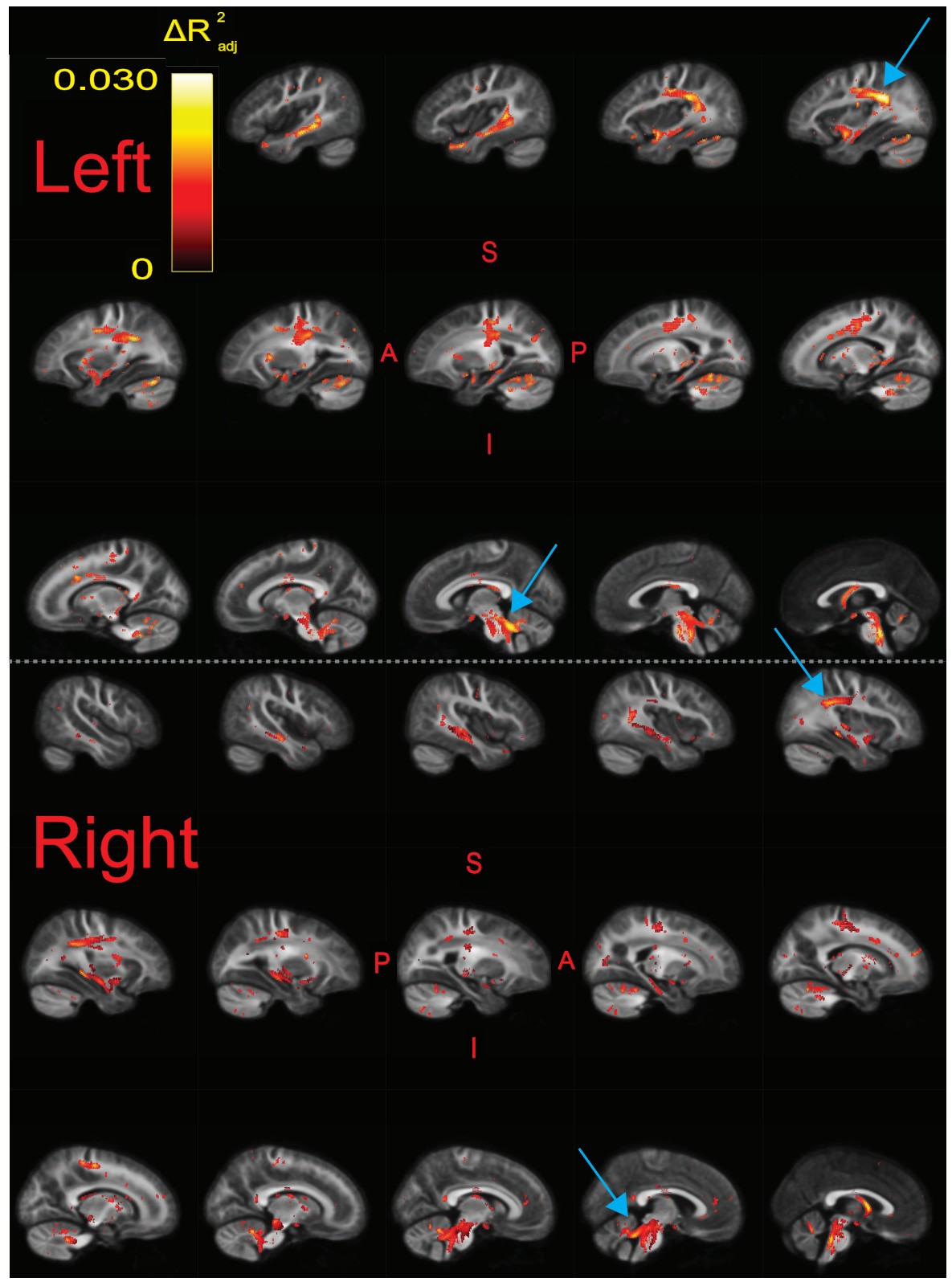

**Figure 3.** Significant fixels ($q_{FDR} < 0.05$) for relating fiber density and cross-section product (FDC) to raw composite Tests of Word Reading Efficiency (TOWRE) scores, colored by effect size ($\Delta R^2_{adj}$). Model confounds included a spline fit for age and linear fits for sex, site, neighbor correlation, and log(ICV). Top and bottom panels are left and right hemispheres, respectively. Sagittal slices go from lateral-to-medial. Blue arrows point to larger

*Figure 3 continued on next page*

*Figure 3 continued*

clusters of fixels in bilateral temporoparietal and cerebellar white matter that were associated with higher effect sizes relative to fixels in the rest of the hemisphere. The template fiber orientation distribution (FOD) image was used as the background image.

The online version of this article includes the following figure supplement(s) for figure 3:

**Figure supplement 1.** Significant fixels ($q_{FDR} < 0.05$) relating fiber density and cross-section product (FDC) to raw composite Tests of Word Reading Efficiency (TOWRE) scores, colored by the beta estimates (top) and direction (bottom; red, LR; green, AP; blue, SI).

**Figure supplement 2.** Plots of the set of tracts in which the strongest effect sizes ($\Delta R^2_{adj} > 0.028$) were achieved for relating fiber density and cross-section product (FDC) to Tests of Word Reading Efficiency (TOWRE) scores (see *Table 2*).

**Figure supplement 3.** Significant fixels ($q_{FDR} < 0.05$) relating fiber cross-section (FC; top), and fiber density (FD; bottom) to raw composite Tests of Word Reading Efficiency (TOWRE) scores, colored by direction (red, LR; green, AP; blue, SI).

**Figure supplement 4.** Significant fixels ($q_{FDR} < 0.05$) relating fiber density and cross-section product (FDC) to raw Sight Word Efficiency (SWE; top) and Phonemic Decoding Efficiency (PDE; bottom) subscores, colored by effect size ($\Delta R^2_{adj}$).

It is encouraging that the fixel-based results highlighted left-hemisphere dorsal temporoparietal white matter as its importance to reading and language has been well-established. The AF and SLF connect inferior frontal and temporoparietal gray matter regions that are essential for language and reading processing (*Catani et al., 2005*). Lesion symptom mapping studies have demonstrated that the AF and SLF are vital connections in the reading network (*Baldo et al., 2018*; *Li et al., 2021*). These tracts, particularly in the left hemisphere, are associated with phonological processing skills (*Yeatman et al., 2011*), which are critical to reading (*Vellutino and Scanlon, 1987*) and impaired in dyslexia (*Swan and Goswami, 1997*). However, the strongest effects in our study were not found in reading-related tracts projecting from the occipital lobe, such as the inferior fronto-occipital fasciculus (IFOF) and inferior longitudinal fasciculus (ILF). Longitudinal studies have suggested that these ventral tracts are more associated with visual orthographic, as opposed to phonological, processing (*Yeatman et al., 2012*; *Vanderauwera et al., 2018*). Our results suggest that phonological skills, as opposed to lower-level visual and orthographic processing, may provide more of a bottleneck to single-word reading abilities in children. The present results are supported by a large-scale longitudinal study finding that FA of the left AF, but not ILF, covaries with single-word reading skill trajectories in children (*Roy et al., 2022*). This notion is also consistent with a behavioral study demonstrating that orthographic skills are more related with the ability to read longer passages as opposed to single words (*Barker et al., 1992*). Thus, FBAs of skills relating to reading longer texts, as opposed to single words, might instead highlight ventral tracts. We also note that the MLF intersected with the significant fixel clusters. This tract has received less attention due to a lack of clear characterization of its structure and function. However, some clinical cases suggest that the left MLF may be associated with verbal-auditory learning and comprehension (*Latini et al., 2021*). We reiterate that the tract masks largely overlapped and should not be used to make definitive associations between fixel-location and bundles, especially because tracts were defined in template, as opposed to native, space.

The present findings suggest that higher FDC in the SCP is associated with better reading skills. Although the cerebellum is not commonly perceived as a core hub in the reading network, theories of reading suggest the cerebellum has a role in fluent word recognition (*Alvarez and Fiez, 2018*; *D'Mello et al., 2020*; *Li et al., 2022*), and cerebellar deficits have been hypothesized as central impairments in dyslexia (*Nicolson et al., 2001*). In particular, the SCP contains efferent fibers that connect deep cerebellar nuclei to contralateral thalamic cortical regions. Co-activation of language-dominant hemispheric inferior frontal regions and contralateral cerebellar regions during verbal tasks (*Jansen et al., 2005*) suggests that the SCP may be a putative tract for cortico-cerebellar interactions in verbal processing. Previous studies have reported that FA of bilateral SCP inversely relates to reading skills (*Travis et al., 2015*; *Bruckert et al., 2020*). We did not find an inverse relationship between FDC and reading abilities, although one should not a priori expect FA and FDC to covary. Despite the lack of a clear consensus of cerebellar contributions to reading abilities, our findings suggest that the cerebellum should remain a focus in studies of reading skills, especially since it is often cropped out of MRI acquisitions.

While the present results suggest a left-sided laterality in FDC-TOWRE correlation effect sizes, it is noteworthy that statistically significant fixels were distributed across the brain. The left-hemispheric laterality is consistent with the frequent focus on predominantly left-sided networks used in reading

**Table 2.** Intersections of white matter tracts with significant fixels for correlations between fiber density and cross-section product (FDC) and reading skill.

The number of fixels is present for two significance thresholds. For tracts that exist bilaterally, results are given in the form of left/right. Tracts in which the maximum effect size ($\Delta R^2_{adj}$) exceeded 0.028 are designated with a bold font. This only happened in the left hemisphere. Tract masks are not mutually exclusive, and nearby tracts likely overlapped to various degrees.

| Tract | N fixels ($q_{FDR} < 0.05$) | N fixels ($q_{FDR} < 0.001$) | Max effect size ($\Delta R^2_{adj}$) |
|---|---|---|---|
| **AF** | 2446/1571 | 186/0 | **0.030**/0.020 |
| ATR | 114/297 | 0/0 | 0.017/0.017 |
| CA | 314 | 2 | 0.018 |
| CC_1 | 53 | 0 | 0.015 |
| CC_2 | 1351 | 0 | 0.018 |
| CC_3 | 197 | 0 | 0.015 |
| CC_4 | 1770 | 3 | 0.021 |
| CC_5 | 1484 | 0 | 0.015 |
| CC_6 | 2022 | 32 | 0.024 |
| CC_7 | 250 | 0 | 0.018 |
| CG | 298/227 | 0/0 | 0.018/0.019 |
| CST | 2561/1789 | 90/109 | 0.024/0.024 |
| FPT | 3171/2809 | 214/221 | 0.024/0.024 |
| FX | 348/300 | 6/6 | 0.024/0.024 |
| ICP | 675/614 | 2/25 | 0.023/0.022 |
| IFOF | 1205/1056 | 26/0 | 0.024/0.018 |
| ILF | 811/422 | 27/0 | 0.021/0.019 |
| MCP | 2043 | 22 | 0.022 |
| **MLF** | 1631/824 | 101/0 | **0.029**/0.020 |
| OR | 585/596 | 18/0 | 0.021/0.016 |
| POPT | 2785/2103 | 118/119 | 0.024/0.021 |
| **SCP** | 1453/1378 | 85/76 | **0.029**/0.021 |
| SLF I | 668/903 | 5/4 | 0.019/0.020 |
| **SLF II** | 918/1015 | 50/0 | **0.029**/0.020 |
| **SLF III** | 741/415 | 116/0 | **0.030**/0.019 |
| ST_FO | 185/125 | 0/0 | 0.019/0.013 |
| ST_OCC | 862/872 | 26/2 | 0.024/0.018 |

*Table 2 continued on next page*

*Table 2 continued*

| Tract | N fixels ($q_{FDR} < 0.05$) | N fixels ($q_{FDR} < 0.001$) | Max effect size ($\Delta R^2_{adj}$) |
|---|---|---|---|
| ST_PAR | 1857/1295 | 15/0 | 0.024/0.020 |
| ST_POSTC | 1463/582 | 9/0 | 0.020/0.016 |
| ST_PREC | 1854/671 | 17/2 | 0.024/0.016 |
| ST_PREF | 825/537 | 0/0 | 0.019/0.017 |
| ST_PREM | 214/95 | 0/0 | 0.019/0.018 |
| STR | 1035/531 | 3/2 | 0.017/0.014 |
| T_OCC | 625/617 | 17/0 | 0.021/0.017 |
| T_PAR | 1436/685 | 4/0 | 0.020/0.016 |
| T_POSTC | 1086/383 | 0/0 | 0.017/0.015 |
| T_PREC | 1497/607 | 5/2 | 0.021/0.014 |
| T_PREF | 748/505 | 0/0 | 0.018/0.017 |
| T_PREM | 51/143 | 0/0 | 0.012/0.014 |
| UF | 665/406 | 23/0 | 0.021/0.016 |

.
AF = arcuate fasciculus; MLF = middle longitudinal fasciculus; SCP = superior cerebellar peduncles; SLF = superior longitudinal fasciculus.
Please refer to Figure 3 of the *TractSeg* publication (**Wasserthal et al., 2018a**) for a full list of the tract abbreviations.

(**Houdé et al., 2010**; **Paulesu et al., 2014**). However, since some theories of dyslexia etiology, such as the anchoring hypothesis (**Ahissar, 2007**) and cerebellar hypothesis (**Alvarez and Fiez, 2018**; **D'Mello et al., 2020**; **Li et al., 2022**), imply that deficits in reading could also arise from domain-general deficits, it is plausible that neural signatures outside of the putative reading network may be informative for predicting reading abilities and disabilities, and that these neural bases are not restrained to properties of white matter. Poor reading abilities have been associated with more global neural differences, most consistently manifested as reductions in intracranial volume (**Ramus et al., 2018**), which we replicate here (**Table 1**, **Figure 2—figure supplement 1**). A functional MRI study found that whole-brain patterns of reading-driven activity conferred advantages to

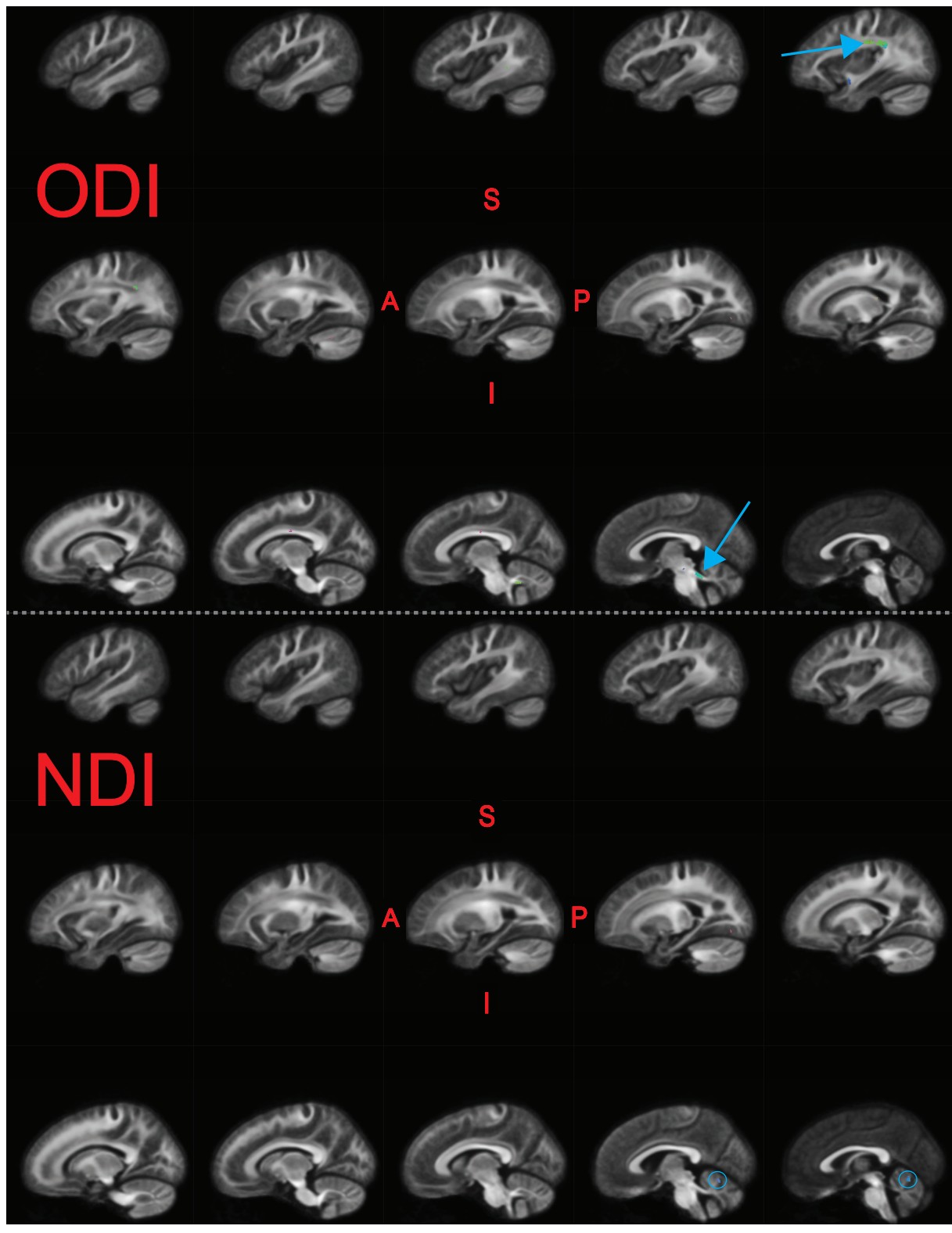

**Figure 4.** Significant fixels ($q_{FDR} < 0.05$) for relating neurite orientation density and dispersion index (NODDI) metrics to raw composite Tests of Word Reading Efficiency (TOWRE) scores, colored by direction (red, LR; green, AP; blue, SI). Model confounds included a spline fit for age and linear fits for sex, site, neighbor correlation, and log(ICV). Top and bottom panels are the indexes for orientation dispersion (ODI) and neurite density (NDI), respectively. Only the left hemisphere is shown. Sagittal slices go from lateral-to-medial. Blue arrows and circles indicate significant fixels. The template fiber orientation distribution (FOD) image was used as the background image.

predicting reading outcomes among dyslexic children compared with targeted region of interest analyses (*Hoeft et al., 2011*). A machine-learning approach to classifying dyslexic from neurotypical children found that white matter features outside of the putative reading network meaningfully improved discriminability (*Cui et al., 2016*). The same group conducted a similar study finding that morphometry of bilateral gray matter regions contributed to predicting continuous reading comprehension scores (*Cui et al., 2018*). Our study adds to these by suggesting that diffuse white matter variation, as indexed by FDC, relates to individual differences in reading abilities independent of ICV (since it was regressed out), although not to a categorical distinction between typical reading ability and reading disability. Future studies should investigate whether multivariate whole-brain patterns of brain morphometry, microstructure, and activity can improve prediction of reading skills, and whether these patterns share biological bases. Such diffuse and multimodal models for predicting reading abilities would likely achieve higher effect sizes than our fixel-specific measures. The changes in $R^2_{adj}$ attributed to the reading measures in predicting fixel metrics were modest, peaking at around 0.030 for the primary analysis, although they are in a similar range of $\Delta R^2_{adj}$ values reported in other brain–behavior correlation studies (e.g., *Pines et al., 2022*).

Although there were significant correlations between single-word reading ability and FDC, there was not an analogous group difference between those with typical reading ability and those with reading disability. This could be in part due to fewer participants being included in the group analyses (total n = 672) compared to the continuous analyses (n = 983). It is also important to consider that collapsing participants into reading proficiency groups loses information about individual differences in reading ability. This could lead to reductions in statistical power if variation in neural metrics truly lies along a spectrum of reading skill. Although it is a worthwhile pursuit to investigate neurodevelopmental bases of dyslexia, which may be addressed by group comparisons, these questions may be better asked in pre-readers based on future reading outcomes (i.e., comparing children who later do and do not develop typical reading skills). Studying pre-readers would help rule out concerns that findings are due to the consequences of developing typical or poor reading skills, as opposed to the etiology, which is a concern for studies of late-stage readers (*Protopapas and Parrila, 2018*; *Protopapas and Parrila, 2019*). There has not yet been a fixel-based analysis focusing on pre-reading skills, but other studies have found white matter microstructural alterations, largely in the left arcuate fasciculus, among pre-readers who have either a familial risk for dyslexia, lower pre-reading skills associated with risk for dyslexia, or future diagnoses of dyslexia (*Saygin et al., 2013*; *Vanderauwera et al., 2015*; *Vandermosten et al., 2015*; *Langer et al., 2017*; *Vanderauwera et al., 2017*; *Wang et al., 2017*; *Yu et al., 2020*).

In our secondary analyses, we replicated recent studies suggesting that FA does not relate to reading skills (*Moreau et al., 2018*; *Koirala et al., 2021*; *Meisler and Gabrieli, 2022*; *Roy et al., 2022*). These null results may arise from the many biological factors that influence FA (*Beaulieu, 2009*; *Johansen-Berg and Behrens, 2013*; *Shemesh, 2018*; *Friedrich et al., 2020*; *Lazari and Lipp, 2021*) and lack of specificity from being defined on the voxel-level (in which crossing fibers may be present), as opposed to fixel-level. Different manifestations of white matter plasticity from learning to read, such as axonal pruning and increased myelination, can have opposing effects on FA (*Yeatman et al., 2012*), confounding FA analyses and their interpretations. The present secondary results, to a limited extent, also replicated *Koirala et al., 2021*, which found negative associations between NODDI metrics and reading skills. The authors attributed this relationship to a more efficient neural architecture. Interestingly, in the present results, areas where ODI negatively related with reading skills approximately overlapped with where FD positively with reading abilities (*Figure 4*, *Figure 3—figure supplement 3*). One may have a priori expected significant FD regions to overlap with those from neurite density (NDI), given both metrics are neural density measures. We can only speculate as to what underlies the observed overlap, and future work should further investigate the relationship between fixel metrics and measures from other DWI models.

Our study contributes to, but still leaves open, the discussion of what properties of brain structure change when developing reading skills. There is a frequent focus on myelin plasticity in learning-driven brain development (*Xin and Chan, 2020*). However, DWI signal is largely insensitive to myelination (*Beaulieu, 2002*). Since this study is cross-sectional, an important unanswered question is whether axonal differences that drive higher FDC are induced by learning how to read, or alternatively whether the presence of higher FDC in putative reading white matter is static and predisposes one to better

reading outcomes. Longitudinal studies of white matter and reading skills have provided some related insights. *Roy et al., 2022* suggests that variance in reading skill over multiple years *precedes* changes in FA. However, *Van Der Auwera et al., 2021* found that lower FA in the left AF among future-dyslexic children existed *prior* to formal reading instruction and predicted future reading scores. The biological basis of FA and its longitudinal change are unclear, and these studies seem to differ regarding the temporal dependence between white matter microstructure and reading skills. At the very least, these studies jointly suggest that white matter is not static in its relation to reading skills. Multimodal studies probing rapid intervention-driven changes have suggested that properties of myelin do not change throughout reading intervention (*Huber et al., 2021*), with mixed evidence for whether MD (which relates to extra-axonal properties) tracks intervention responses (*Huber et al., 2018*; *Partanen et al., 2021*). Future work will need to be done to distinguish age-related from learning-related development in different time scales and to ascribe these changes to biophysical phenomena, which are nontrivial tasks (*Jelescu et al., 2020*). *Liang et al., 2021* demonstrated that fixel metrics can undergo even rapid plasticity. Thus, we hope future studies of reading will use longitudinal FBA (*Genc et al., 2018*) to investigate long-term and rapid reading-driven plasticity. FDC from the fixel-based analysis is more specific than FA, conveying information about intra-axonal volume on the fixel-level (*Dhollander et al., 2021a*). However, one should still interpret FDC findings cautiously. DWI alone cannot discern whether intra-axonal volume is driven by the number of axons or width of axons, and fixel-based metrics have not yet been validated against histological standards.

Our findings contribute to a growing list of cross-sectional studies suggesting that models more nuanced than the diffusion tensor better capture variance in reading skills (*Zhao et al., 2016*; *Koirala et al., 2021*; *Sihvonen et al., 2021*; *Economou et al., 2022*). Unlike many prior studies, we ran a whole-brain analysis instead of running statistics on metrics averaged within tracts. This has important implications for interpreting results. Our whole-brain findings suggest a relationship between reading skills and FDC in fixel-specific regions shared across participants. However, this does not preclude the possibility of tract-averaged diffusion metrics relating to reading skills, even among areas that yielded few significant fixels. A disruption in white matter leading to a deficit in reading might happen at any location along a tract, and variance in such locations across participants could lead to null findings on a fixel-by-fixel level. Whole-brain analyses are also prone to stricter correction for multiple tests. On the other hand, the spatial specificity achieved by whole-brain FBAs could be informative for speculating about the outcomes of white matter disruptions. White matter bundles do not only deliver signals from one end to the other; they branch off and synapse at multiple locations along its course. Thus, spatially specific disruptions of signal could have different downstream effects, warranting a more nuanced approach. Considering that tracts such as the AF and SLF have distinct cores that subserve reading and math processing (*Grotheer et al., 2019*), averaging over an entire tract may introduce noise by considering parts of the bundle that are not relevant to the behavior being studied. However, one can functionally localize white matter tracts by finding the streamlines that connect participant-specific reading functional regions (*Grotheer et al., 2022*). One can also extract tract-wise measures similar to FDC that relate to the intra-axonal volume of the bundles (*Smith et al., 2022*). This approach may lead to appreciable insights into properties of long-range connections that underlie reading skills with higher and more interpretable effect sizes.

In our previous work (*Meisler and Gabrieli, 2022*), we correlated diffusion metrics with each TOWRE subtest score individually. However, in this study, we used the composite TOWRE measure as the primary phenotypic variable of interest. Our rationale in doing so is the same as in *Sihvonen et al., 2021*: a composite score is more stable as it is more robust to variance due to temporary attention lapses, which may only affect performance on one test. In addition, running fewer models mitigates the problem of multiple hypothesis testing. We acknowledge, however, that real word and pseudoword reading may rely on different skills. Pseudoword reading ability, for example, is considered a more pure gauge of phonological processing skills because the novelty of these nonwords precludes one from relying on memorized representations. We share the model results for relating FDC to SWE and PDE scores in the supplementary materials (*Figure 3—figure supplement 4*). The two maps were qualitatively similar to the model results for the composite TOWRE measure. This is not entirely surprising given the high degree of correlation between the subscores (*Figure 2*).

Our model outputs also allowed us to visualize the impact that intracranial volume and image quality had on DWI-derived metrics. Recent studies relating ICV (*Eikenes et al., 2022*) and image

quality (*Koirala et al., 2022*) to DTI measures have important implications for model specification that should be extended to fiber-specific metrics in future work. For each metric across fixel-based, DTI, DKI, and NODDI models, we found diffuse significant correlations with both ICV and neighbor correlation (with the exception of FD, since ICV was not part of the model). The nature of these associations varied between the different DWI metrics and covariates, and a full characterization of these relationships falls outside the scope of this report. However, we encourage interested readers to visualize these associations using the model outputs we shared (see 'Data and code availability') and to consider including these metrics in their own fixel-based analysis models. It should be emphasized that our models include multiple predictors that may covary with ICV and image quality, or otherwise not be of interest if one wanted to rigorously characterize the effects of ICV or image quality. However, future work should comprehensively characterize and explain the impacts of brain volume and image quality on diffusion-weighted signal.

The present findings should be interpreted in the context of several other limitations. First, it was not made available what specific criteria were used to diagnose reading disabilities. This is why we used stringent criteria based on clinical and reading assessments to define the RD group. Secondly, most participants in the HBN present with at least one psychological, learning, or neurodevelopmental disorder (*Alexander et al., 2017*). The diversity of the cohort, while perhaps more representative of a population, presents multiple phenotypic factors that could confound results. To maintain high statistical power and a diverse sample, we did not exclude participants based on the presence of other neurodevelopmental or learning disorders such as ADHD or specific language impairments. Such co-occurring difficulties occur at high rates in reading disorders; for example, approximately 50% of children with reading disorders also qualify for a diagnosis of ADHD (*Willcutt et al., 2010*; *DuPaul et al., 2013*; *Al Dahhan et al., 2022*). Exclusion of such co-occurring difficulties would yield a nonrepresentative sample of those with reading disability.

Further, since white matter bundles can have different shapes across participants (*Yeatman et al., 2011*; *Wassermann et al., 2011*) and analyses are performed in a single template space, an effect in a region of fixels could be partially driven by global geometric variations across participants. Similarly, the fixel-to-tract attributions should be cautiously interpreted since our tracts were delineated on the FOD template of 38 participants, and tract segmentations tend to overlap (*Schilling et al., 2022*). The b-value of 2000 s/mm$^2$, while higher than the b-value of typical DTI acquisitions, is not exceptionally large compared to the spectrum of values typically employed in FBA. Thus, our measures of FD, and therefore FDC as well, may have been partially undermined by contamination from extra-axonal signal (*Genc et al., 2020*). Finally, we reemphasize that our study is cross-sectional and correlational. Thus, it cannot be used to make causal conclusions of white matter's contributions to reading skills. We hope our work will inform future fixel-based investigations using longitudinal, mediation, modeling, or prediction approaches that can warrant stronger claims.

## Conclusion

In this study, we examined whether fixel-based metrics from 983 children and adolescents covaried with single-word reading abilities or were reduced among those with reading disabilities. We found that higher FDC related to better single-word reading abilities, but that FDC did not differ significantly between children with and without reading disabilities. The strongest associations between FDC and reading aptitude were localized in left-hemisphere temporoparietal and cerebellar white matter, which is consistent with prior neuroanatomical studies of reading and literacy. The fixel-based analysis is a promising approach to investigating reading in future studies, capturing variance in reading skill when multiple other DWI-derived scalars failed to do so, and parameters of DWI acquisitions should be considered with this in mind.

## Materials and methods
### Participants

We downloaded preprocessed DWI and phenotypic data from 2136 participants across the first eight data releases of the HBN project (*Alexander et al., 2017*). Phenotypic data were accessed in accordance with a data use agreement provided by the Child Mind Institute. Preprocessed DWI data were provided as part of the HBN Preprocessed Open Diffusion Derivatives (HBN-POD2) dataset

(*Richie-Halford et al., 2022*). The HBN project was approved by the Chesapeake Institutional Review Board (now called Advarra, Inc; https://www.advarra.com/, protocol number: Pro00012309). Informed consent was obtained from all participants ages 18 or older. For younger participants, written informed consent was collected from their legal guardians, and written assent was obtained from the participants. Detailed inclusion and exclusion criteria for the HBN dataset are described in the project's publication (*Alexander et al., 2017*). Of note, each participant was fluent in English, had an IQ > 66, and did not have any physical or mental disorder precluding them from completing the full battery of scanning and behavioral examinations.

Several behavioral and cognitive evaluations were collected as part of HBN. Relevant to this study, participants completed the Test of Word Reading Efficiency 2nd edition (TOWRE; *Torgesen et al., 1999*). The TOWRE consists of two subtests, Sight Word Efficiency (SWE) and Phonemic Decoding Efficiency (PDE). For these tests, each participant is shown a list of either real words (SWE) or pronounceable nonwords/pseudowords (PDE) and is then asked to read the items aloud as quickly as possible. Raw scores are based on the number of items read correctly within the 45 s time limit and are then converted to an age-standardized score (population mean = 100, standard deviation = 15). A composite standardized TOWRE score is calculated as the mean of the standardized PDE and SWE scores. Most participants also completed the Edinburgh Handedness Inventory (EHI; *Oldfield, 1971*), Barratt Simplified Measure of Social Status (BSMSS; *Barratt, 2006*), and Wechsler Intelligence Scale for Children 5th edition (WISC; *Wechsler and Kodama, 1949*).

After quality control (see 'Data inclusion and quality control'), there were 983 participants ages 6–18 years old. We divided these participants into two groups based on diagnostic criteria and standardized reading scores (*Figure 2*). A total of 102 participants were diagnosed with a 'specific learning disability with impairment in reading; following the 5th edition of the Diagnostic and Statistical Manual for Mental Disorders (*Edition, 2013*) and scored ≤ 85 on both TOWRE subtests (age-standardized). These participants were placed in the RD group. A total of 570 participants who were not diagnosed with a reading impairment and scored ≥ 90 on both TOWRE subtests (age-standardized) were placed in the TR group. The remaining 311 participants were not placed into either group, but were still included in the correlation analyses across all participants.

## Neuroimaging acquisition

Detailed scanner protocols for each site are published on the HBN project website (http://fcon_1000.projects.nitrc.org/indi/cmi_healthy_brain_network/File/mri/). Data were collected using either a 1.5T Siemens mobile scanner (Staten Island site) or a 3T Siemens MRI scanner (sites at Rutgers University Brain Imaging Center, Cornell Brain Imaging Center, and the City University of New York Advanced Science Research Center). All participants were scanned while wearing a standard Siemens 32-channel head coil. A high-resolution T1-weighted (T1w) image was collected for all participants, with parameters that slightly varied between sites. A DKI scan was acquired with 1.8 mm isotropic voxel resolution, 1 b = 0 s/mm$^2$ image, and 64 noncollinear directions collected at b = 1000 s/mm$^2$ and b = 2000 s/mm$^2$. A pair of PEpolar fieldmaps were collected before the diffusion scan to quantify magnetic field susceptibility distortions.

## Neuroimaging minimal preprocessing

Minimally preprocessed data were downloaded from HBN-POD2 and produced by *QSIPrep* (*Cieslak et al., 2021*) 0.12.1 (https://qsiprep.readthedocs.io/en/latest/), which is based on *Nipype* 1.5.1 (*Gorgolewski et al., 2011*; *Gorgolewski et al., 2018*) (RRID:SCR_002502). Many internal operations of *QSIPrep* use *Nilearn* 0.6.2 (*Abraham et al., 2014*) (RRID:SCR_001362) and *Dipy* (*Garyfallidis et al., 2014*). The following two sections contain text from boilerplates distributed by *QSIPrep* under a CC0 license with the expressed intention of being incorporated into manuscripts for transparency and reproducibility. We made minor changes for succinctness and completeness.

### Anatomical preprocessing

The T1w image was corrected for intensity nonuniformity (INU) using N4BiasField Correction (*Tustison et al., 2010*) (ANTs 2.3.1) and used as T1w-reference throughout the workflow. The T1w-reference was then skull-stripped using antsBrainExtraction.sh (ANTs 2.3.1) using OASIS as target template. Brain tissue segmentation of CSF, white matter (WM), and gray matter (GM) was performed on the

brain-extracted T1w using FAST (*Zhang et al., 2001*) (FSL 6.0.3:b862cdd5, RRID:SCR_002823). Additionally, in order to calculate intracranial volumes, we ran recon-all (*FreeSurfer* 6.0.1, RRID:SCR_001847; *Dale et al., 1999*; *Buckner et al., 2004*; *Fischl, 2012*) as part of *sMRIPrep* 0.8.1 (*Esteban et al., 2021*) to reconstruct brain surfaces.

## Diffusion image preprocessing

Denoising using dwidenoise (*Veraart et al., 2016*) was applied with settings based on developer recommendations. Gibbs unringing was performed using *MRtrix3*'s mrdegibbs (*Kellner et al., 2016*). Following unringing, B1 field inhomogeneity was corrected using dwibiascorrect from *MRtrix3* with the N4 algorithm (*Tustison et al., 2010*). After B1 bias correction, the mean intensity of the DWI series was adjusted so all the mean intensity of the b = 0 images matched across each separate DWI scanning sequence. *FSL*'s (version 6.0.3:b862cdd5) eddy was used for head motion correction and Eddy current correction (*Andersson and Sotiropoulos, 2016*). eddy was configured with a $q$-space smoothing factor of 10, a total of five iterations, and 1000 voxels used to estimate hyperparameters. A linear first-level model and a linear second-level model were used to characterize Eddy current-related spatial distortion. $q$-space coordinates were forcefully assigned to shells. Field offset was attempted to be separated from participant movement. Shells were aligned post-eddy. eddy's outlier replacement was run (*Andersson et al., 2016*). Data were grouped by slice, only including values from slices determined to contain at least 250 intracerebral voxels. Groups deviating by more than 4 standard deviations from the prediction had their data replaced with imputed values. Here, b = 0 fieldmap images with reversed phase-encoding directions were used along with an equal number of b = 0 images extracted from the DWI scans. From these pairs the susceptibility-induced off-resonance field was estimated using a method similar to that described in *Andersson et al., 2003*. The fieldmaps were ultimately incorporated into the Eddy current and head motion correction interpolation. Final interpolation was performed using the jac method. The preprocessed DWI time series were resampled to ACPC, and their corresponding gradient directions were rotated accordingly.

## Fixel-based analyses (FBA)

### Fixel metric calculations

Comprehensive details of this workflow have been described elsewhere (*Raffelt et al., 2012b*). Preprocessed DWI volumes and brain masks were reoriented to the *FSL* standard orientation. The gradient table was correspondingly rotated with *MRtrix3*'s dwigradcheck. We then upsampled the DWI image and brain masks to 1.25 mm isotropic voxels. We extracted only the highest diffusion shell (b = 2000 s/mm², along with the b = 0 volumes) to proceed with estimating the constrained spherical deconvolution (CSD) fiber response functions and FODs, as to limit the influence of extra-axonal signal (*Genc et al., 2020*). Response functions for white matter, gray matter, and CSF were estimated with *MRtrix3*'s unsupervised dhollander algorithm (*Dhollander et al., 2016*; *Dhollander et al., 2019*). For each tissue compartment, site-specific average fiber response functions were calculated across participants (*Raffelt et al., 2012b*), which enable valid inter-subject comparisons while controlling for scanner differences across sites (*Smith et al., 2022*). Participant FODs for each tissue compartment were calculated using Single-Shell 3-Tissue CSD (SS3T-CSD) (*Dhollander and Connelly, 2016*) from *MRtrix3Tissue* (https://3Tissue.github.io), a fork of *MRtrix3* (*Tournier et al., 2019*). FODs were normalized using log-domain intensity normalization (*Raffelt et al., 2017a*; *Dhollander et al., 2021b*).

We then generated an unbiased study-specific FOD template and warped individual participant FOD images to this template (*Raffelt et al., 2011*; *Raffelt et al., 2012a*). Due to the large size of our participant cohort, we could not feasibly use all FOD images to generate a population template. To decide which participants were used to inform the template, we divided the age range of participants into 10 uniformly spaced bins. In each age bin, we selected two males and two females. Within sex groupings, the participant in the TR and RD group with the highest quality control prediction score ('XGB score,' see *Richie-Halford et al., 2022*) was selected to be in the template. There were no females in the RD group among the two oldest age bins, so our template was composed of 38 participants. We implemented this method to make a robust high-quality template that was unbiased by sex and included representation from a wide range of ages and reading levels.

Participant FOD images were registered to template space. The same transformation was used to warp brain masks to template space. A whole-brain template-space analysis mask was calculated

as the intersection of all participants' warped masks, such that each region would contain data from all participants. Within this voxel-wise template mask, a whole-brain fixel-wise analysis mask was segmented from the FOD template. Participant fixels were segmented from their warped FODs (*Smith et al., 2013*), and then reoriented and mapped to the template space. Fiber density (FD) was calculated for each fixel by taking the integral of its corresponding FOD lobes (*Raffelt et al., 2012b*). Fiber cross-sections (FC) were also calculated for each fixel, informed by the geometric distortions needed to warp from native-to-template space (*Raffelt et al., 2017b*). The product of FD and FC was also calculated (FDC) (*Raffelt et al., 2017b*). We applied a log transform to FC so that it would be normally distributed and centered around 0. FDC was calculated before this log transformation was applied.

A whole-brain tractogram with 20 million streamlines was generated from the FOD template using seeds uniformly distributed across the template-space voxel-wise mask (*Tournier et al., 2010*). SIFT filtering (*Smith et al., 2013*) was applied to account for false positives in streamline generation (*Maier-Hein et al., 2017*), resulting in a pruned tractogram with 2 million streamlines. This was used to create a fixel-to-fixel connectivity matrix. This connectivity data was used to inform spatial smoothing of FD, log(FC), and FDC maps, such that smoothing at a given fixel only occurred within that fixel's fiber population, thus mitigating partial-volume effects or influences from crossing fibers (*Raffelt et al., 2015*).

## Tract segmentation

We extracted the three primary spherical harmonic peaks of the template FOD image within the voxel-wise brain mask (*Jeurissen et al., 2013*). These peaks were input to *TractSeg* 2.3 (*Wasserthal et al., 2018a*; *Wasserthal et al., 2018b*; *Wasserthal et al., 2019*), a convolutional neural network-based tract segmentation and reconstruction pipeline that strikes a favorable balance between the subjectivity of manual delineation and objectivity of automated atlas-based tracking approaches (*Genc et al., 2020*). We created tractograms for all 72 fiber-bundles produced by *TractSeg*. We generated 10,000 streamlines per tract (up from the default of 2000) to reduce inter-run variability from the stochastic nature of reconstruction. From each set of fiber bundle streamlines, we created a corresponding tract fixel density map, which we binarized to create tract fixel masks.

## Statistics

We considered a diverse set of potential confounds to include in our statistical models. These included age (*Genc et al., 2018*; *Dimond et al., 2020*), sex (*Lyon et al., 2019*; *Kirkovski et al., 2020*), handedness (*Honnedevasthana Arun et al., 2021*), socioeconomic status (SES) as indexed by the average years of parental education from the BSMSS, visuospatial IQ index from the WISC (*Ramus et al., 2018*), globally averaged fixel metrics (gFD, gFC), log-transformed intracranial volume (ICV) (*Smith et al., 2019*), and scanning site (*Schilling et al., 2021b*). We also considered multiple quality covariates, including mean framewise displacement, and neighbor correlation (*Yeh et al., 2019*). The machine-learning-based quality score distribution from *Richie-Halford et al., 2022* was skewed towards 1 and not normally distributed, and thus was not a good candidate confound. Since gFD and gFC are calculated within fixels, and fixels are only segmented in white matter, differences in white matter volumetric proportions should not influence global fixel metrics. As exploratory analyses, we ran Spearman correlations between all continuous variables to inform our decision of model covariates and look for well-established trends in behavioral and neuroimaging metrics, validating the data collection procedures (*Figure 2—figure supplement 1*).

To run our statistical models, we used *ModelArray* 0.1.2 (*Zhao et al., 2022*). This R-based software package minimizes memory consumption to allow analysis of all participants and enables GAM on fixel data, which is especially useful for cohorts with a wide age range (*Bethlehem et al., 2022*). We ran two models for our primary analyses: a regression of FDC against the raw TOWRE composite score, and a comparison of FDC between the TR and RD groups. We restricted our primary analyses to FDC based on recent guidance surrounding the control of false positives in FBA (*Smith et al., 2021*), but we also ran analogous models for FD and log(FC) to explore the contributions of fiber microstructure and morphometry in a *post hoc* fashion. Model confounds included a smooth penalized spline fit for age (maximum of four inflection points) and linear fits for sex, site, quality (neighbor correlation), and log(ICV). Log(ICV) was not included as a covariate for models of FD (*Smith et al.,*

*2019*). Categorical variables (group, sex, and site) were coded as factors, and continuous variables (TOWRE scores, neighbor correlation, age, and ICV) were mean-centered and rescaled to unit variance to mitigate concerns of multicollinearity and poor design matrix conditioning. Effect sizes for the predictors of interest (TOWRE score or group label) were calculated as the difference in adjusted $R^2$ coefficients ($\Delta R^2_{adj}$) between the full statistical model fit and the fit of a reduced model without the primary predictor variable (TOWRE scores or group label). p-values were corrected across the brain using Benjamini–Hochberg FDR correction (*Benjamini and Hochberg, 1995*). To ascribe significant fixels to tracts, we intersected significant fixels ($q_{FDR} < 0.05$) and the binarized tract fixel masks. We note that tract masks tended to overlap (*Schilling et al., 2022*), so a single fixel could be associated with multiple fiber bundles.

Given the wide age range of participants, we additionally explored whether the relationship between FDC and reading skills varied with age. We ran a smooth bivariate interaction model, which can gauge whether there is an interaction between two continuous variables accounting for nonlinear effects (*Wood, 2017*). This model included the same linear confounds as the main FDC model, but had smooth terms for age, raw composite TOWRE scores, and the interaction between the two. These splines were unpenalized tensor product smooth terms.

## Fitting and analysis of DTI, DKI, and NODDI models

As additional exploratory analyses, we also ran models relating reading abilities with scalar maps from diffusion tensor models, diffusion kurtosis models, and NODDI models. We used *QSIPrep* version 0.15.3 to run the dipy_dki (*Henriques et al., 2021*) and amico_noddi (*Daducci et al., 2015*) reconstruction pipelines on the preprocessed data. From the dipy_dki pipeline, we collected FA, MD, KFA, and MK. From amico_noddi, we collected the NDI (synonymous with ICVF) and ODI. We resampled and warped these scalar maps to the 1.25 mm isotropic template space, and then mapped the voxel values to fixels. While each fixel in a voxel was initially assigned the same value, spatial smoothing was still applied on the fiber population level. We then used *ModelArray* to run models relating each of these metrics to the composite raw TOWRE scores. Similar to the primary analyses of FDC, model confounds included a penalized spline fit for age and linear fits for sex, site, quality (neighbor correlation), and log(ICV).

## Data inclusion and quality control

We downloaded preprocessed DWI (*Richie-Halford et al., 2022*) and phenotypic data from 2136 participants across the first eight data releases of the HBN project (*Alexander et al., 2017*). HBN-POD2 distributes a quality metric accompanying each image that predicts the probability that the image would pass manual expert quality review ('xgb_qc_score', or 'dl_qc_score' if the former score was not available) (*Richie-Halford et al., 2022*). It ranges from 0 (no chance of passing expert review) to 1 (image will definitely pass expert review). We excluded any participants with a quality score of less than 0.5. Twenty different DWI acquisition parameters were present across participants (*Covitz et al., 2022*; *Richie-Halford et al., 2022*). We only included participants who had images acquired with the most common acquisition parameters in their site ('SITE_64dir_most_common'). We also excluded any participant who (1) was outside ages 6–18; (2) had missing basic demographic or TOWRE scores; or (3) failed *FreeSurfer* reconstruction. Based on these criteria, 986 participants advanced to the fixel-based analysis. Fiber response functions could not be obtained for two of these participants due to nonpositive tissue balance factors. After registering the participant FODs to the template FOD, we overlaid each participant's registered brain mask on top of the registered FOD image as a quality control check that registration was successful. This revealed one participant with an unsuccessful registration to template space who was excluded from analyses. Therefore, a total of n = 983 participants (570 TR, 102 RD, 311 other) passed all quality control procedures and were included in subsequent analyses.

## Data and code availability

Preprocessed neuroimaging data can be downloaded following directions from the HBN-POD2 manuscript (*Richie-Halford et al., 2022*), and phenotypic data can be collected following directions on the HBN data portal (http://fcon_1000.projects.nitrc.org/indi/cmi_healthy_brain_network/index.html) after signing a data use agreement. All instructions and code for further processing data and running the statistical models can be found at https://github.com/smeisler/Meisler_Reading_

FBA (copy archived at swh:1:rev:aefac140776bd0f04ac4abae38e6458a7cf7ec27) (*Meisler, 2022*). With minimal modification, the neuroimaging processing code should be able to run on most BIDS-compliant datasets using the SLURM job scheduler (*Yoo et al., 2003*). The HBN data use agreement precludes us from sharing model inputs since they contain restricted phenotypic data. However, we share the population FOD template, tract segmentations, and model outputs (which only report data in the aggregate) at https://osf.io/3ady4/. These can all be viewed using *MRview* from *MRtrix3*. Some software we used were distributed as Docker (*Merkel, 2014*) containers, then compiled and run with Singularity 3.9.5 (*Kurtzer et al., 2017*):

- *QSIPrep* 0.15.3 (singularity build qsiprep.simg docker://pennbbl/qsiprep:0.15.3)
- *TractSeg* 2.3 (singularity build tractseg.simg docker://wasserth/tractseg:master)
- *MRtrix3* 3.0.3 (singularity build mrtrix.simg docker://mrtrix3/mrtrix3:3.0.3)
- *MRtrix3Tissue* 5.2.9 (singularity build mrtrix3t.simg docker://kaitj/mrtrix3tissue:v5.2.9)
- *sMRIPrep* 0.8.1 (singularity build smriprep.simg docker://nipreps/smriprep:0.8.1)
- *FSL* 6.0.4 (singularity build fsl.simg docker://brainlife/fsl:6.0.4-patched)
- *ModelArray* 0.1.2 (singularity build modelarray.simg docker://pennlinc/modelarray_confixel:0.1.2)

We encourage anyone to use the latest stable releases of these software.

## Acknowledgements

We thank the Child Mind Institute for their diligence in collecting and sharing the neuroimaging and behavioral data and the authors of the HBN-POD2 manuscript for sharing the preprocessed derivatives. We thank all of the participants and their families for volunteering their time to be involved in the Healthy Brain Network. We thank Chenying Zhao, Matt Cieslak, and Theodore Satterthwaite for developing and guiding the use of the *ModelArray* software. This work was supported by the National Institute on Deafness and Other Communication Disorders (NIDCD) (grant numbers 5T32DC000038-29 and 5T32DC000038-30), the Halis Family Foundation, and Reach Every Reader, a grant supported by the Chan Zuckerberg Foundation.

## Additional information

### Funding

| Funder | Grant reference number | Author |
|---|---|---|
| National Institute on Deafness and Other Communication Disorders | 5T32DC000038 | Steven Lee Meisler |
| Chan Zuckerberg Initiative | | John DE Gabrieli |

The funders had no role in study design, data collection and interpretation, or the decision to submit the work for publication.

### Author contributions

Steven Lee Meisler, Conceptualization, Software, Formal analysis, Investigation, Visualization, Methodology, Writing – original draft, Writing – review and editing; John DE Gabrieli, Supervision, Funding acquisition, Writing – original draft, Writing – review and editing

### Author ORCIDs

Steven Lee Meisler http://orcid.org/0000-0002-8888-1572

### Ethics

Human subjects: The Healthy Brain Network project was approved by the Chesapeake Institutional Review Board (now called Advarra, Inc.; https://www.advarra.com/; protocol number: Pro00012309). Informed consent was obtained from all participants ages 18 or older. For younger participants, written informed consent was collected from their legal guardians, and written assent was obtained from the participants.

Decision letter and Author response
Decision letter https://doi.org/10.7554/eLife.82088.sa1
Author response https://doi.org/10.7554/eLife.82088.sa2

---

# Additional files

## Supplementary files

• MDAR checklist

• Transparent reporting form

• Supplementary file 1. ANOVA results for site-wise comparisons between phenotypic and neuroimaging metrics. Group comparison columns list significant $t$-statistics. *p<0.05 for the ANOVA between all sites. Post hoc $t$-tests were only run if the between-sites ANOVA was significant. Only significant $t$-statistics (p<0.05) are shown in the table. A positive $t$-statistic denotes Site 1 > Site 2. EHI, Edinburgh Handedness Inventory; SES, socioeconomic status; ICV, intracranial volume; TOWRE, Tests of Word Reading Efficiency composite score, age-normalized; WISC VSI, Wechsler Intelligence Scale for Children visuospatial index, age-normalized; WISC VCI, Wechsler Intelligence Scale for Children verbal comprehension index, age-normalized; gFD, globally averaged fiber density; gFC, globally averaged fiber cross-section.

## Data availability

Raw and preprocessed neuroimaging data from the Healthy Brain Network (*Alexander et al., 2017*) are publicly available without restriction, and can be downloaded from Amazon Simple Storage Service (S3) using Amazon Web Services tools following directions from the HBN-POD2 manuscript (*Richie-Halford et al., 2022*). Raw neuroimaging data may also be downloaded directly from the Healthy Brain Network data portal (http://fcon_1000.projects.nitrc.org/indi/cmi_healthy_brain_network/sharing_neuro.html#Direct%20Down). Access to full Healthy Brain Network phenotypic and behavioral data, which are stored at https://data.healthybrainnetwork.org/main.php, is restricted. For this reason, we cannot make our full study outputs publicly available. These data can be collected by any entity for non-commercial purposes following directions on the Healthy Brain Network data portal (http://fcon_1000.projects.nitrc.org/indi/cmi_healthy_brain_network/Pheno_Access.html) after signing a data use agreement. Study-specific code and instructions for processing data and running the statistical models can be found at https://github.com/smeisler/Meisler_Reading_FBA (copy archived at swh:1:rev:aefac140776bd0f04ac4abae38e6458a7cf7ec27). We share the population FOD template, tract segmentations, and model outputs (which only report data in the aggregate) at https://osf.io/3ady4/. These can all be viewed using MRview from MRtrix3.

The following dataset was generated:

| Author(s) | Year | Dataset title | Dataset URL | Database and Identifier |
|---|---|---|---|---|
| Meisler S | 2022 | Fixel Based Analyses of Reading Abilities | https://doi.org/10.17605/OSF.IO/3ADY4 | OSF, 10.17605/OSF.IO/3ADY4 |

---

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
