## [Editor Report]

This valuable study investigates the association between fixel-based white matter measures and reading for the first time. In a large sample of participants ranging from 6-18 years of age, a convincing association between intra-axonal volume and single-word reading abilities are reported. This work will be of interest to a wide readership.

---

## [Decision Letter]

**Decision letter after peer review:**

Thank you for submitting your article "Fiber-Specific Structural Properties Relate to Reading Skills in Children" for consideration by *eLife*. Your article has been reviewed by 2 peer reviewers, and the evaluation has been overseen by a Reviewing Editor and Timothy Behrens as the Senior Editor. The following individual involved in review of your submission has agreed to reveal their identity: Kathryn Y Manning (Reviewer #1).

Essential revision:

1. I would recommend using raw (not age-corrected) measures for both the brain (FDC) and the behavioral level (reading scores) and control for age in the statistical analyses. It would also be interesting to examine whether the correlation changes across development.

2. P 12 line 259 "one should not a priori expect Fa and FDC to covary". Indeed, we cannot assume this based on the lack of studies investigating the link between FA and FDC. Therefore, I would recommend adding analyses that directly link FA and FDC, using the same method (such as tract-based analyses like in Meisler & Gabrieli, 2022).

3. The effect sizes are rather small, more specifically, in the cluster of fixels with the largest effect sizes, the change in R square was only around 3%. Hence this indicates that only a very small portion of the reading variance is explained by the brain measures. This should be discussed.

4. In the results the focus seems to be on the FDC-measure, and only post-hoc analyses of FD and FC. However, at the end of the introduction, it seems that all three measures will be primary variables of interest ('fixel-based measures", line 145). This is important because there are currently no corrections for the fact that 3 measures and not 1 primary measure (FDC) was used. Only in the Methods (line 524) it becomes clear that the primary analyses is using FDC, but this Method-section comes after Introduction, Results and Discussion. Is FDC a more standard measure than FD and FC in previous studies? Also, does 'post-hoc exploration' means that FD and FC are only investigated when FDC is significant? Because for group analyses, differences in FD and FC are investigated despite the absence of a significant group effect for FDC.

---

## [Author Response]

Essential revision:

1. I would recommend using raw (not age-corrected) measures for both the brain (FDC) and the behavioral level (reading scores) and control for age in the statistical analyses. It would also be interesting to examine whether the correlation changes across development.

We thank the reviewer for raising this issue. We have rerun the correlation analyzes using a composite raw score calculated as the sum of the raw PDE and SWE scores, as well as with the individual raw subscores (as suggested in your 6th comment). Qualitatively, the main result has not changed: there are still diffuse regions of “statistically significant” positive associations of FDC with reading, with highest effect sizes achieved in left temporoparietal and cerebellar regions. The effect sizes (changes in adjusted R^2^ values) were slightly smaller when using the raw scores (peaked at ~0.30 as opposed to the original peak of ~0.34). As further clarification, we did not make any change to our group analyses, which use standardized scores to classify RD vs. TR readers.

To test whether the correlations were stable across development, we ran an additive bivariate smooth interaction model. These are used to gauge the significance of the interaction between two continuous variables when nonlinear effects are expected. The formula used to run this was as follows (using R syntax conventions):

FDC ~ ti(AGE, k = 4, fx = TRUE) + SEX + SITE + ICV + N_CORR + ti(TOWRE, fx = TRUE, k = 4) + ti(AGE, TOWRE, k = 4, fx = TRUE)

The last term is the one whose significance relates to whether correlation changed across age. The interaction term was not significant in any of the fixels that showed a significant FDC-TOWRE relationship. Of note, the fx = TRUE terms indicate that the spline fits for the AGE and TOWRE terms are unpenalized, which is more appropriate when one wants to perform significance testing on the smooth terms. In every other analysis, the age spline is penalized, which provides a better fit at the expense of significance testing validity for that term.

2. P 12 line 259 "one should not a priori expect Fa and FDC to covary". Indeed, we cannot assume this based on the lack of studies investigating the link between FA and FDC. Therefore, I would recommend adding analyses that directly link FA and FDC, using the same method (such as tract-based analyses like in Meisler & Gabrieli, 2022).

We agree that more research should be done to investigate relationships between measures from different DWI models. Some work has already been done on DTI and NODDI (e.g., Huber et al., 2019; *Developmental Cognitive Neuroscience*). However, we have a few reasons to suggest that such an analysis as suggested would fall outside the scope of this manuscript:

1) The HBN data we use is not ideal for this purpose. The Human Connectome Project has a larger sample size, as well as more directions at both lower b-values (higher SNR, better for calculating FA and other tensor-based metrics) and higher b-values (better for resolving spherical harmonics / fixel metrics).

2) Without histological measures or other maps of interest (such as myelin maps from myelin water imaging of T1/T2 ratios), and without longitudinal neuroimaging data, it would be hard to draw meaningful conclusions about the biological bases of such relationships.

3) This finding would not directly lend itself to reading skills, which is the main focus of this manuscript.

4) FA is defined on the voxel-level whereas FDC is a fixel-level metric. It is also not clear whether one would expect linear vs. nonlinear relationships between metrics. This makes a direct comparison more difficult, and the methods to do so properly may have to be given more thought. This is discussed more in depth in Dhollander et al., 2021; *NeuroImage*.

3. The effect sizes are rather small, more specifically, in the cluster of fixels with the largest effect sizes, the change in R square was only around 3%. Hence this indicates that only a very small portion of the reading variance is explained by the brain measures. This should be discussed.

We thank the reviewer for this comment. We now note this in the discussion (lines 330-333), and explain that these effect sizes are in similar ranges to effect sizes observed in other brain-behavior correlations (e.g., Pines et al., 2022; *Nature Communications*). In addition, as we explain in the discussion, one may not expect fixel-wise measures, which are spatially specific and limited to white matter, to explain a large amount of variance in reading skills, as reading skills may be associated with more global neural variance that is not limited to white matter. This is elaborated more in response to the 9th comment from this reviewer (discussing the local vs. global findings).

4. In the results the focus seems to be on the FDC-measure, and only post-hoc analyses of FD and FC. However, at the end of the introduction, it seems that all three measures will be primary variables of interest ('fixel-based measures", line 145). This is important because there are currently no corrections for the fact that 3 measures and not 1 primary measure (FDC) was used. Only in the Methods (line 524) it becomes clear that the primary analyses is using FDC, but this Method-section comes after Introduction, Results and Discussion. Is FDC a more standard measure than FD and FC in previous studies? Also, does 'post-hoc exploration' means that FD and FC are only investigated when FDC is significant? Because for group analyses, differences in FD and FC are investigated despite the absence of a significant group effect for FDC.

We thank the reviewer for allowing us to clarify this. There is recent guidance (Smith et al., 2021; *ISMRM* conference submission), mentioned in the *Statistics* section of Methods and Materials, describing that primary statistical inferences should be performed on FDC, with post-hoc analysis of FD / FC as warranted. Since there are multiple years between the introduction of fixel-based analysis and publication of this guidance, several prior fixel-based analyses did not take this approach. Nevertheless, we believe focusing on FDC as the primary measure is warranted. We now clarify throughout the manuscript that the primary analysis and hypothesis are focused on FDC. We also remove FD and FC results for the group analysis given that FDC did not yield significant fixels.